# The Structural Characteristics of Seaweed Polysaccharides and Their Application in Gel Drug Delivery Systems

**DOI:** 10.3390/md18120658

**Published:** 2020-12-21

**Authors:** Haowei Zhong, Xiaoran Gao, Cui Cheng, Chun Liu, Qiaowen Wang, Xiao Han

**Affiliations:** College of Biological Science and Engineering, Fuzhou University, Fuzhou 350108, China; N195720010@fzu.edu.cn (H.Z.); N185720009@fzu.edu.cn (X.G.); ibptlc@fzu.edu.cn (C.L.); wangqiaowen8@163.com (Q.W.)

**Keywords:** alginic acid, carrageenan, seaweed polysaccharides, gels, drug delivery

## Abstract

In recent years, researchers across various fields have shown a keen interest in the exploitation of biocompatible natural polymer materials, especially the development and application of seaweed polysaccharides. Seaweed polysaccharides are a multi-component mixture composed of one or more monosaccharides, which have the functions of being anti-virus, anti-tumor, anti-mutation, anti-radiation and enhancing immunity. These biological activities allow them to be applied in various controllable and sustained anti-inflammatory and anticancer drug delivery systems, such as seaweed polysaccharide-based nanoparticles, microspheres and gels, etc. This review summarizes the advantages of alginic acid, carrageenan and other seaweed polysaccharides, and focuses on their application in gel drug delivery systems (such as nanogels, microgels and hydrogels). In addition, recent literature reports and applications of seaweed polysaccharides are also discussed.

## 1. Introduction

Marine plants are mainly lower algae plants, higher plants and seed plants. Based on their pigment, cell structure, reproductive method and reproductive organ structure, we generally divide marine algae plants into: Cyanophyta, Chlorophyta, Xanthophyta, Euglenophyta, Pyrrophyta, Chrysophyta, Charophyta, Rhodophyta, Bacillariophyta and Phaeophyta. Among these, Rhodophyta, Phaeophyta and Chlorophyta are the most widely used in biomaterials. With the recent development and progress in scientific research, polysaccharides, polyphenols, amino acids, polyunsaturated fatty acids, terpenes and other bioactive components have been isolated from seaweed. These compounds exhibit excellent anti-inflammatory, antioxidant, bacteriostatic, anticancer and anticoagulant effects [1,2,3,4,5,6], which makes the use of algae in food, medical and other fields increasingly prominent.

In the utilization of algae resources, algae polysaccharides are an extremely important component. Algae polysaccharides are a multi-component mixture composed of one or more monosaccharides, which are connected with 1, 3 and 1, 4 glycosidic bonds, distributed between and within the algae cells. They are generally water-soluble, and some contain sulfate groups, which provide high viscosity or solidification properties. There are many types of seaweed polysaccharides that arise from different sources, which are divided into Rhodophyta polysaccharides, Fucophyta polysaccharides, Chlorophyta polysaccharides, etc. [2]. The main types of algae and their products are shown in Figure 1.

In recent years, more in-depth research on the medicinal value of algae polysaccharides has been conducted, which has attempted to establish the mechanism behind the medicinal functions of algae polysaccharides both in vivo and in vitro, as well as promoted the development and utilization of algae polysaccharides in pharmaceutical preparations. Wang [7] reported that seaweed polysaccharides have good inhibitory effects on matrix metalloproteinases (MMPs), which improves the upregulation of MMP expression caused by ultraviolet b (UVB) radiation, inflammatory cytokines and certain hormones and chemical agents, as well as reducing the inflammation and degradation of connective tissue caused by free radicals and reactive oxygen species (ROS) initiating the mitogen-activated protein kinase pathway [8]. In addition, it showed a good promoting effect on reducing the damage that is induced by free radicals and ROS [9]. Other reports have summarized the potential anticancer mechanisms of algal polysaccharides, including induction of apoptosis, cell cycle arrest, regulation of transduction signal pathways, inhibition of migration and angiogenesis and activation of the immune response and antioxidant system [10,11,12]. They found that vascular endothelial growth factor/vascular endothelial growth factor receptor 2 (VEGF/VEGFR2), transforming growth factor β receptor/Smad/Snail (TGFR/Smad/Snail), Toll-like receptor4/reactive oxygen species/endoplasmic reticulum (TLR4/ROS/ER), chemokines CXC ligand 12/CXC receptor 4 (CXCL12/CXCR4), TGFR/Smad7/Smurf2, phosphatidylinositol 3-kinase/protein kinase B/the mammalian target of rapamycin (PI3K/AKT/mTOR), PDZ-binding kinase /t-lymphokine-activated killer cell-originated protein kinase (PBK/TOPK) and β-catenin/Wnt are the main cellular signaling pathways that play a key role in the prevention and treatment of tumorigenesis using algal polysaccharides [12]. With the continuous development of biomedical materials, research into new types of natural materials has received the most attention. Seaweed polysaccharides have physical structural properties and natural biological activity, and they can be extensively sourced at low cost. They hold promise for further application in various controllable and sustained anti-inflammatory and anticancer drug delivery systems, such as seaweed polysaccharide-based nanoparticles, microspheres and gels, etc. Here, gel drug delivery systems (including nanogels, microgels and hydrogels) have internal cross-linked structures that can load various drugs, thereby providing a wide range of applications in the field of drug delivery.

In order to discuss the application of seaweed polysaccharides in gel drug delivery in recent years, we searched the relevant literatures of seaweed polysaccharides based gel drug delivery system in the past three years through the websites of pubmed, Google academic and school library electronic journal platform, etc. They were divided into two categories: reviews and articles. From the reviews, we summarized the properties and characteristics of polysaccharides, and traced back the valuable citations. From the articles, the studies focusing on gel drug delivery system are extracted and summarized. Through sorting out the literature information, three types of seaweed polysaccharides were selected according to the sources of polysaccharides: alginic acid and its derivatives, carrageenan and its derivatives and other seaweed polysaccharides. Hence, in this review, the structural characteristics of the above three types of seaweed polysaccharides are summarized and the latest progress in the application of seaweed polysaccharide-based gels in drug delivery systems is discussed. We aim to provide a reference for the future research and development of seaweed polysaccharides. The summary of this review is shown in Figure 2, and the details will be expanded later.

## 2. Alginic Acid and Its Derivatives

Alginic acid, also known as acidum alginic, is essentially a straight chain block glucuronic acid, which is widely found in hundreds of brown algae, Phaeophyta. In its natural state, it combines with various cations in seawater to form various alginates, which exist in the cell wall and play a strengthening role. In the process of extraction, it can be converted into sodium, potassium, ammonium, calcium salt or other derivatives, which we call algin; commercial algin is mainly sodium alginate. As alginic acid and its derivatives have good stability, solubility and viscosity and are safe, they are often used as excipients for food or pharmaceutical preparations to achieve thickening, stability and emulsification [13]. However, recent studies have shown that alginic acid and its derivatives also have good biocompatibility, antioxidant activity and anti-inflammatory activity [14,15,16,17]. At present, they are mostly commonly applied in cosmetics, and there are various anti-wrinkle, moisturizing and anti-ultraviolet products available [6]. Application in drug delivery systems is recent and demonstrates considerable application potential, which will drive the direction of future development.

### 2.1. Structure and Characteristics

The alginic acid product is a white to yellowish brown powder, the average molecular weight is approximately 240,000, and its melting point is above 300 °C. Insoluble in cold water and organic solvents, slightly soluble in hot water, its aqueous solution viscosity is approximately five times that of starch. It is resistant to acid, but decarboxylation occurs when it is reacted with concentrated hydrochloric acid. It exhibits selective adsorption effects towards metal ions [16], in particular, Fe (Ⅱ) ions [18,19].

From a structural point of view, alginic acid is a copolymer composed of β-d-manuronic acid (mannuronate, M) and amurl guronic acid (guluronate, G) connected with β-1,4 glycosidic bonds and randomly arranged into poly-GG, poly-MG and poly-MM fragments [20], as shown in Figure 3. The linear polymers contain carboxyl groups in each uronic acid unit, and exhibit clear pH sensitivity [21]. When alginic acid is in the alkali metal salt and ammonium salt state, the solution is neutral or alkaline, and at the same time a high proportion of carboxylate groups (–COO^−^)result in polyanion electrolyte properties, meaning the aqueous solution has certain adhesion properties. When the pH value increases, the carboxylic acid group (–COOH) continuously dissociates, the molecular chain extends and the solubility increases. However, under acidic conditions, the conversion of –COO^−^ to –COOH is not as facile, the solubility decreases and the molecular chain shrinks. Below pH 3, the alginic acid solution is converted into insoluble alginic acid gel or a precipitate, which is accelerated by the addition of multivalent metal ions (such as calcium chloride) [22].

### 2.2. Sodium Alginate Gels

Sodium alginate can quickly form a gel under extremely mild conditions, and this gel-forming property is directly related to the G and M content. In the presence of Ca^2+^, Sr^2+^ and other cations [23,24], the Na^+^ on the G unit reacts with the cation and the G unit accumulates to form a crosslinked network structure, resulting in the formation of hydrogel. The conditions required for this gel formation and dissociation process are very mild, which effectively avoid the inactivation of sensitive drugs, proteins, cells and enzymes, resulting in excellent biocompatibility and drug loading properties.

The gel formed from sodium alginate and calcium ions is resistant to freezing and can be recovered using water absorption after drying. The gel strength is directly affected by the concentration of calcium ions and sodium alginate. The higher the concentration, the greater the strength of the gel. The gelation process can be controlled by adjusting the pH, selecting a suitable calcium salt, adding phosphate buffer or chelating agent, or by the gradual release of polyvalent cations or hydrogen ions, or both [25]. The viscosity of sodium alginate is closely related to gel brittleness; the higher the viscosity, the more brittle the gel. By adjusting the ratio of sodium alginate to acid, the rigidity of the gel can be adjusted. Similarly, by controlling the solubility of the calcium salt, the type and rigidity of the gel can be adjusted. A gel can be made quickly using soluble calcium chloride [26]. When calcium dihydrogen phosphate is used, calcium is only released when the temperature rises to between 93 and 107 °C, which delays the gelation time. When the addition of calcium ions reaches 2.3%, a thick gel is obtained; when calcium ion addition is less than 1%, it is a mobile body. When the pH approaches the isoelectric point of the protein, the protein and sodium alginate form a soluble complex, and the viscosity increases, which inhibits protein precipitation; when the pH decreases further, the complex precipitates [16].

### 2.3. Application of Sodium Alginate-Based Gels in Drug Delivery Systems

By analyzing the structure and reaction characteristics of alginic acid and its derivatives, it has been found that gels based on sodium alginate have excellent potential for application in various drug delivery systems. The pH-responsive gel property allows adaptation to acid-base changes in the microenvironment, such as in the digestive system. Lin SY et al. [27] prepared a series of calcium alginate particles and dispersed 5-aminosalicylic acid onto the surface of the particles using spraying. Next, different glue coating layers were embedded (Aquacoat and Eudragit L-30D). When the external pH changes, the calcium alginate particles swell under the action of osmosis, and gradually exceed the loading strength of the gel coating. In this case, the tablets disintegrate and the drug is dispersed allowing slow and controlled release. Similarly, Nilay Kahya et al. prepared sodium alginate (NaAlg)/sodium carboxymethyl cellulose (NaCMC) composite hydrogel beads by crosslinking with a barium chloride solution to carry methotrexate [28]. T.S. Anirudhan et al. developed a novel pH-sensitive bioactive amine mesoporous silica-alginate/folic acid conjugated o-carboxymethyl chitosan–gelatin (AMSN-Alg/FA-CMCT-Gel) composite nanogel system, which was used to transport 5-Fluorouracil (5-FU) and didemethoxycurcumin (BDC). The combination of nano-encapsulation and BDC improved the efficiency of 5-FU in the treatment of colorectal cancer [29]. With the development of nano-biomaterials, these properties have been increasingly applied in microstructure technology [16,30,31,32,33,34,35,36].

In addition, the temperature responsive characteristics of alginate gels provide numerous possibilities for the use of alginate gels as intelligent drug carriers. Durkut S et al. developed a thermosensitive poly(*N*-vinylcaprolactam)-grafted aminated alginate (PNVCL-g-Alg-NH_2_), which underwent a phase transition and water absorption expansion at approximately 35 °C, which is suitable for the physiological temperature environment. Furthermore, copolymerization with PNVCL reduces the water absorption of the aminated alginate and improves its thermal stability. In terms of biocompatibility, in vitro cytotoxicity and blood compatibility analyses confirmed that the PNVCL-g-Alg-NH_2_ scaffolds were not cytotoxic and did not induce hemolysis [37]. Min Liu et al. coupled poly(*N*-isopropylacrylamide) (PNIPAAm) with sodium alginate to synthesize a thermally responsive copolymer alginate-g-PNIPAAm (Figure 4) [38]. The copolymer was dissolved in water or a phosphate buffered saline solution at room temperature (25 °C). When the temperature rose above the critical micelle temperature, a self-assembled micelle with a low critical micelle concentration was formed, and when the temperature rose to body temperature (37 °C), the copolymer transformed into a thermosensitive hydrogel. To demonstrate a practical application, doxorubicin (DOX) was loaded into the sodium alginate-g-PNIPAAm to construct an injectable thermally-responsive sustained release hydrogel. The continuous release of DOX-encapsulated micelles effectively enhanced the DOX uptake of multidrug resistant AT3B-1 cells and was effective at killing cancer cells, demonstrating that drug resistance was overcome.

## 3. Carrageenan and Its Derivatives

Carrageenan is a high molecular weight hydrophilic polysaccharide extracted from red algae, Rhodophyta. It is one of the most widely used algin products in industrial applications, along with agar, in the world. It is a translucent sheet or powder with a white to yellowish brown appearance, has a wrinkled surface and is slightly glossy, odorless and tasteless, sticky and slippery. It is often used in the food industry, such as during the manufacture of jelly, ice cream, canned food, etc., and has good water retention, thickening, emulsifying and gelling characteristics, as well as being safe and non-toxic.

### 3.1. Structure and Characteristics

Carrageenan is a linear polysaccharide compound formed from alternating sulfated or non-sulfated galactose units and linked with α-1,3glycosidic bond and β-1,4-galactose bonds. It can carry a sulfate group on the D-galactose unit C-4′ connected at the 1-position and has a molecular weight of more than 200,000. It can be derived from the calcium, potassium, sodium and ammonium salts of the polysaccharides sulfates and composed of various galactose and dehydrated galactose units. In addition, depending on the position at which the semi-ester sulfate group is connected on the galactose unit, it can be divided into seven main types: κ-carrageenan, ι-carrageenan, λ-carrageenan (Figure 5), γ-carrageenan, ν-carrageenan, ξ-carrageenan and μ-carrageenan [39]. The κ-type carrageenan forms a brittle gel with potassium ions and forms a soft, elastic gel on interaction with calcium ions; λ-carrageenan does not form a gel when combined with a salt. Generally, the carrageenan produced from algae is not a pure version of one of the above products, but exists in a mixture.

The sulfate radical on the carrageenan molecule has a strong negative charge and adheres to polycationic macromolecules and amphoteric proteins. Below the isoelectric point, the electrostatic action resulting from a positive and negative charge produces precipitation; above the isoelectric point, the two hold the same charge, and there are polyvalent cations acting as a binding agent with carrageenan to form a hydrophilic colloid; at the isoelectric point, precipitation occurs via the combination of polyvalent cations with carrageenan. These properties are beneficial for the encapsulation and release of proteins, and allow combination with polycations to form new carriers for delivery systems.

Although carrageenan has many natural advantages and has been widely used in many cosmetics and foods without any adverse effects, the subcutaneous administration of carrageenan has resulted in paw edema in mice. Therefore, it is an inflammatory stimulant of the anti-inflammatory experimental mouse model, and means there is a risk of immunological complications during systemic administration, which limits its application in drug delivery systems. Fortunately, in recent years, it has been found that the biocompatibility of carrageenan derivatives can be greatly improved using structural modification. Chemically modified carrageenan was reviewed in detail by Zia et al. [40]. It was found that the biofunctional properties of carrageenan depend, to a large extent, on sulfate groups and their substitution. The addition of sulfate groups makes carrageenan possess both biological activity and biocompatibility, making it more suitable for drug delivery. Moreover, carrageenan exhibits additional properties following modification. When studying the rheology of oxidized red algal galactan, Vanina et al. found that agarose and κ-carrageenan were oxidized by (2,2,6,6-tetramethylpiperidinyl) oxy (TEMPO) in the presence of NaOCl and NaBr; the resulting products, which had several oxidation states, were then characterized. These low oxidation state derivatives are sensitive to heat as well as pH and calcium, similar to natural polysaccharides [41], which results in a corresponding response and release when introduced into complex and changeable microenvironments in the body.

### 3.2. Carrageenan-Based Gels

The properties of carrageenan gels are related to its chemical composition, structure and molecular size. The formation of carrageenan gels can be divided into four stages: when dissolved in hot water, the molecule is irregularly curled; when the temperature decreases, the molecule forms a single helix; when the temperature drops further, a double helix is formed between the molecules (a three-dimensional network structure), and solidification begins to occur; when the temperature drops even further, the double helix aggregates to form a gel. The formation of the gel is also affected by the presence of cations, and the existence of Ca^2+^, and K^+^ further reduces the helix–helix aggregation induced by intermolecular interactions, forming an ordered three-dimensional network [42]. κ-carrageenan exhibits univalent and divalent cation-dependent gelation behavior. However, in the presence of these cations, λ-carrageenan does not form a gel, but the viscosity of the solution increases. The absence of a 3,6-anhydro bridge unit in λ-carrageenan and the inward orientation of the sulfate group at the 2 position prevent crosslinking of the adjacent double helix, thus preventing gel formation [43]. The dual response to different ions and temperature means the gel has excellent potential in drug delivery systems.

Further, the gel state of carrageenan is also affected by the acidity and basicity, with viscosity and gel strength lost at pH 4.3 and when heated. This is due to the hydrolysis of carrageenan at low pH, which disconnects the bonds within 3,6-dehydration-d-galactose. The degree of hydrolysis increases at higher temperatures and lower cation concentrations. However, once the temperature of the solution is lower than the gel temperature, potassium ions bind to the sulfate group on carrageenan, which prevents hydrolysis.

### 3.3. Application of Carrageenan-Based Gels in Drug Delivery Systems

Following the development and progress made with respect to carrageenan modification, a number of drug delivery breakthroughs have been achieved. Sato et al. prepared temperature-responsive polysaccharide particles through the formation of an emulsion and the subsequent sol–gel transformation of polysaccharides. It was found that sunken κ-carrageenan particles may remain in the alveoli, and were likely to produce significant therapeutic effects in pulmonary administration [44].

Malairaj Sathuvan et al. successfully prepared pH-responsive κ-Car-Cur active drug carriers using solvent volatilization and freeze-drying methods. As shown in Figure 6, when adapted to the tumor microenvironment, the drug release was more active when at pH 5.0, and the cumulative release rate reached 78%. When applied in vitro, and compared with free curcumin (CUR), the κ-Car-Cur drug-loaded complex induced apoptosis of A549 tumor cells more effectively [45]. Furthermore, Latufa Youssouf et al. used protection/de-protection technology to transplant a polycaprolactone (PCL) chain onto oligo-carrageenan to obtain a polycaprolactone-grafted carrageenan. The amphiphilic polymer formed a spherical nanogel with an average size of 187 ± 21 nm, which could effectively encapsulate hydrophobic drugs such as curcumin and release them over a period of 24–72 h [46]. Azizi, S et al. prepared crosslinked carrageenan/silver nanoparticle hydrogel beads using κ-carrageenan and biosynthetic silver nanoparticles (Ag-NPs) as the matrix. Compared with a pure κ-carrageenan hydrogel, the biological nanocomposite hydrogel exhibited less swelling behavior. It also demonstrated good antibacterial activity against *Staphylococcus aureus*, methicillin resistant *Staphylococcus aureus*, *Pseudomonas aeruginosa* and *Escherichia coli*, with a maximum inhibitory interval of 11 mm. The cytotoxicity study showed that, at a concentration of less than 1000 μg/mL, the non-toxic biological nanocomposite hydrogel had great pharmacological potential [47].

## 4. Other Seaweed Polysaccharides

### 4.1. Structure and Characteristics of Other Seaweed Polysaccharides

Many types of seaweed polysaccharide have been shown to exhibit value in terms of their structural and biological properties; these include fucoidan and β-glucan extracted from Rhodophyta, porphyrin and sulfated galactan extracted from Phaeophycophyta, Ulvans, xylan and mannan extracted from Chlorophyta and GA3P extracted from marine microalgae.

Fucoidans is a polysaccharide from *Saccharomyces cerevisiae*, is mainly derived from brown algae, and contains fucose and sulfate groups. The analysis of fucoidan isolated from kelp indicated it was composed of 97.8% fucose and 2.2% galactose. The analysis of glycosidic bonds showed that (1,3)-α-1-ethyl fuco-pyranose (31.9%) was the main residue, followed by 1,2-linked (13.2%) and 1,4-linked (7.7%) fucose and highly branched (22.4%). A large number of studies have shown that fucoidan induces cytotoxicity towards various types of cancer cell, induces apoptosis and inhibits the invasion, metastasis and angiogenesis of cancer cells [48,49,50,51,52,53,54,55,56,57,58]. Damonte et al. discussed the structure of alginate in detail along with its biological functions in the mechanisms of apoptosis, invasion, metastasis, angiogenesis and the growth and signaling of cancer cell growth, which proved the medicinal value of these alginoglycans [59]. Recently, researchers have studied six types of fucoidan (FRF) including FRF F0.7, F1.5 and F2.1, and found they can be used as antioxidants to protect bone tissue from oxidative stress and may represent an adjuvant for the treatment of bone fragility by counteracting oxidation [60].

β-glucan is a polysaccharide connected by β-glucose residues through various glycosidic bonds. In previous studies, it was mainly found in oats, barley and yeast and more recently has been found in some brown algae. The formation of a linear polysaccharide chain is obtained via the hydrolysis of kelp protease, and is therefore also known as kelp polysaccharide. Lee et al. [61] described the potential wound healing, immune regulation and anti-tumor activity of kelp polysaccharides. It was reported that kelp polysaccharides have regulatory effects including anti-systemic inflammation via reduction of inflammatory cell recruitment in the liver and reduction of inflammatory mediator expression. Kelp polysaccharides purified from Irish brown seaweed (*Laminaria hyperborea*) and kelp have significant antioxidant, radical scavenging for 1-diphenyl-2-picrylhydrazine (DPPH) and antibacterial activity [62]. According to Tsiapali et al. [63], *Laminaria* polysaccharides have strong oxygen free radical absorption abilities, with an EC50 value of 460 mg/mL. It is speculated that the anti-inflammatory effects may be related to the resistance to the oxidation of β(1→3)-glucan. Although kelp polysaccharides do not form hydrogels in their natural state, photocrosslinked laminarine chemically modified using glycidyl methacrylate exhibits hydrogel properties as well as the ability to encapsulate human adipose-derived stem cells [64].

Ulvan is an acidic structural polysaccharide, which exists in the cell walls of green algae (*Ulva lactuca* and *Enteromorpha*). It is composed mainly of rhamnose 3-sulfate, xylose 2-sulfate, glucuronic acid and iduronic acid, and belongs to a group of highly sulfated polysaccharides. Studies have shown that Ulvan exhibits a variety of therapeutic activities, including antibacterial, immunostimulatory, anti-tumor, antioxidant, anti-hyperlipidemia, antiviral and anticoagulant (Table 1). The polysaccharide chain of Ulvan is unstable and is affected by the growth environment and the extraction methods used. More specifically, temperature and the ionic environment seriously affect the structure and morphology of Ulvan, as well as the properties of its gels. For example, under alkaline conditions, Ulvan forms a gel in the presence of Ca^2+^ and B^+^ ions, with the existence of divalent cations enhancing the gelation ability, leading to the formation of a thermally reversible gel. As well as being pH-, ion- and temperature- responsive, the repeating unit within Ulvan has chemical affinity with glycosaminoglycans (such as hyaluronic acid and chondroitin sulfate); therefore, it should also have potential application value in gel drug delivery systems [65].

GA3P is composed of the marine microalgae *Dinoflagellate gymnodinium* SP. The structure of the extracellular polysaccharide is mainly d-galactosome sulfated with L-lactic acid. However, GA3P exhibits strong cytotoxicity regardless of whether lactic acid was present. It can induce apoptosis of human leukemia K562 cells. GA3P is also a strong contact-reactive inhibitor of topoisomerase Ⅰ and Ⅱ and the mechanism of its inhibitory effect is similar to that of dextran sulfate. In contrast to camptothecin (CPT) and teniposide (VM-26), the inhibitory effect of GA3P on topoisomerase Ⅰ and Ⅱ does not involve the accumulation of the DNA topoisomerase Ⅰ and Ⅱ cleavage complex; that is to say, it does not have topoisomerase toxicity, but is a catalytic inhibitor with dual activity. When GA3P was added to a reaction mixture containing CPT or VM-26, it inhibited the stability of the fission complex induced by CPT or VM-26 [90].

The biological and pharmacological activities of several typical marine polysaccharides are shown in Table 1 [66,67,68,69,70,71,72,73,74,75,76,77,78,79,80,81,82,83,84,85,86].

### 4.2. Application of Other Seaweed Polysaccharides Based Gels in Drug Delivery Systems

In recent years, in addition to alginic acid and carrageenan, the other common seaweed polysaccharides have been industrially produced on a large scale, and many attempts have been made to apply different seaweed polysaccharides in gel drug delivery systems. Chetna Verma et al. prepared cisplatin (CP)-xanthane gum (TG)-lecithin (LC) nanogels via a nano-emulsion method. These nanogels had a polygonal core–shell structure, in which the CP particles were embedded in the TG core of the nanogel and covered in the form of an LC shell. To study the drug release mechanism, different concentrations of CP were added to the nanogel; the results showed that the nanogel achieved sustained drug release (Figure 7) [91]. Khan H, et al. prepared superabsorbent hydrogels (CMA-g-PAm) using anionic, cold water-soluble carboxymethyl agarose (CMA, a seaweed polysaccharide derivative) and polyacrylamide (PAm) through rapid microwave assisted grafting. Then, they prepared an anionic carboxymethyl agarose-responsive intelligent super absorbent hydrogel based on anionic carboxymethyl agarose, which was able to carry and release the anticancer drug DOX in a controlled manner [92]. Feki An et al. prepared a sensitive chitosan-red marine algae composite polysaccharide hydrogel and selected the protein drug insulin as the model drug to test the in vitro release behavior of the hydrogel; the composite was based on blue crab chitosan (Cs) and red marine macroalgae *Falkenbergia rufolanosa* polysaccharide (FRP). The structure, morphology, thermal and antioxidant properties of the FRP/Cs hydrogels were characterized. The data showed that the addition of FRP enhances water retention, water absorption and texture. In addition, as shown by the swelling ratio test, the hydrogels were sensitive to pH, ionic strength and temperature. In addition, compared with the samples immersed in pH 7.4 PBS, the hydrogel degradation in pH 1.2 phosphate buffered saline was higher. Similarly, the kinetics of insulin release from the FRP/Cs hydrogels indicated higher insulin release in acidic systems. This study demonstrated that the FRP/Cs hydrogels provide a suitable and promising microenvironment for drug delivery [93].

## 5. Summary and Outlook

In conclusion, among the wide variety of seaweed polysaccharides, this review is focused on alginic acid and its derivatives in brown algae, carrageenan and its derivatives in red algae, and other kinds of seaweed polysaccharides. Their value is not only limited to anti-inflammatory, bacteriostatic, anticancer and other medicinal values, but also extends to the important potential value in gel drug delivery systems. For example, due to the influence of Ca^2+^ and Na^+^ concentration within the plasma, the strength of hydrogels prepared by alginic acid and its derivatives extracted from brown algae increases with increasing cation concentration. And a large number of carboxyl groups make it pH-responsive, so the adaptive hydrogel particles can be applied in the controlled release of drugs in gastrointestinal and tumor microenvironments. Similarly, modified carrageenan is temperature- and pH-responsive. These characteristics are across all seaweed polysaccharides, and the gel particles prepared from them exhibit both intelligent responses and biocompatibility. To summarize, ionic conditions, temperature and pH are the main factors that influence changes in the properties of the gels prepared by seaweed polysaccharides. These properties can be utilized to design gels with intelligent response mechanisms, such as gel films, gel microspheres, nanogels and other intelligent drug delivery gels, and may be applied in the delivery of a wide variety of drugs.

## Figures and Tables

**Figure 1 marinedrugs-18-00658-f001:**
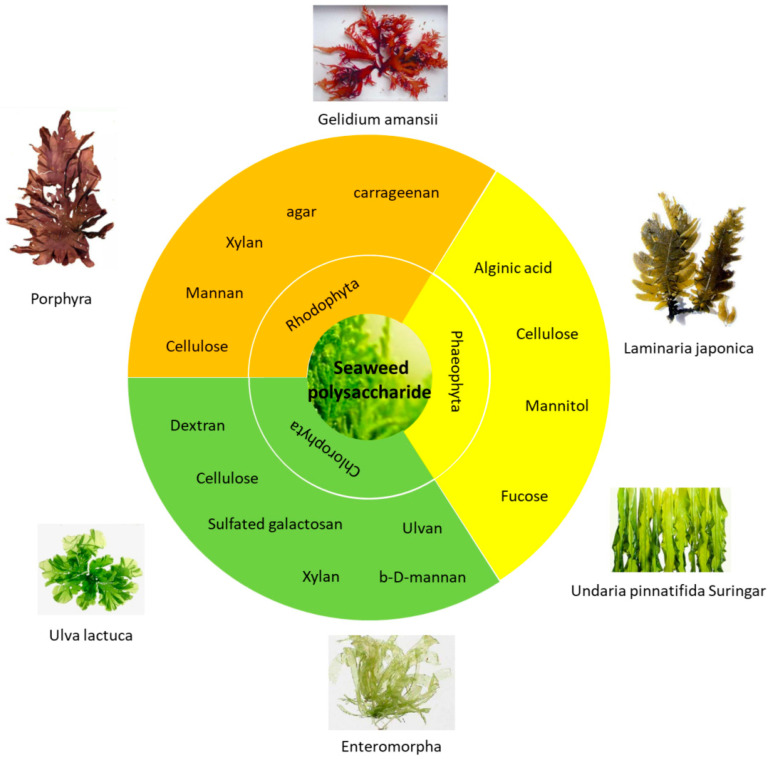
Classification of the main seaweed polysaccharides and their sources.

**Figure 2 marinedrugs-18-00658-f002:**
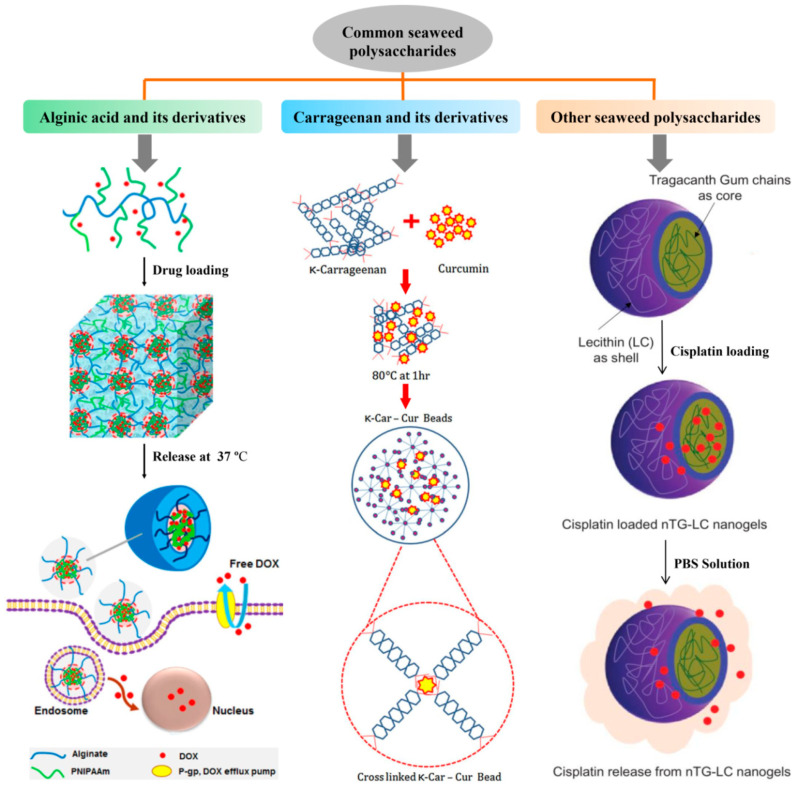
Structural characteristics of seaweed polysaccharides and their application in nano/micro/hydro-gel drug delivery systems.

**Figure 3 marinedrugs-18-00658-f003:**
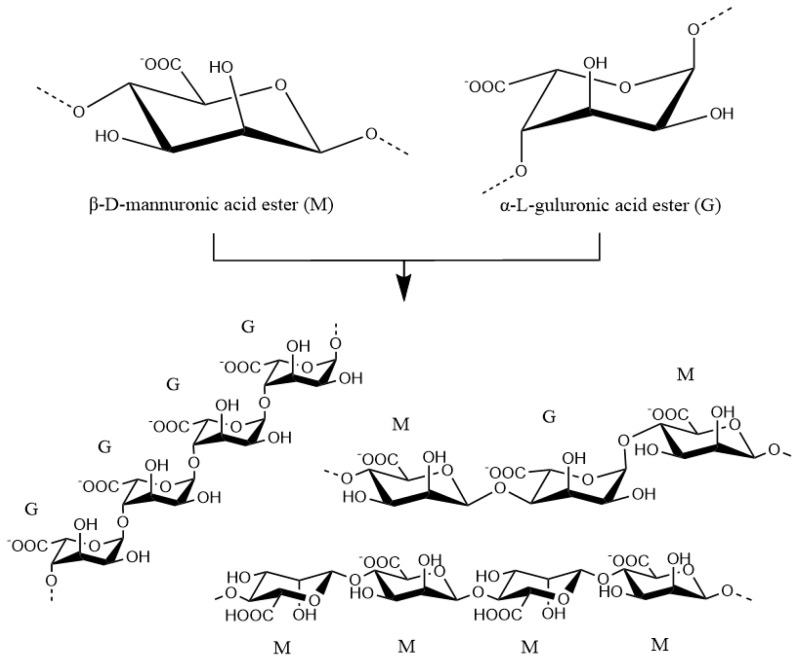
β-d-mannuronic acid ester (M) and α-l-guluronic acid ester (G) are connected by β-1,4 glycosidic bonds to form poly GG, poly MG, poly MM fragments, and then form alginic acid Copolymer.

**Figure 4 marinedrugs-18-00658-f004:**
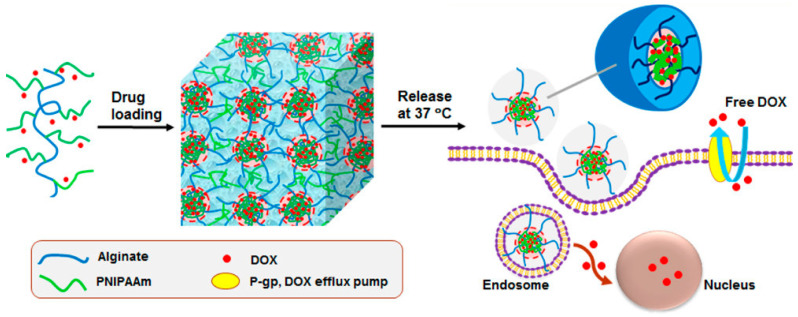
The technology route of thermal response copolymer sodium alginate-g-poly (*N*-isopropylacrylamide) (alginate-g-PNIPAAm), Reproduced with permission from [38]. ACS Appl. Mater. Interfaces, 2017.

**Figure 5 marinedrugs-18-00658-f005:**
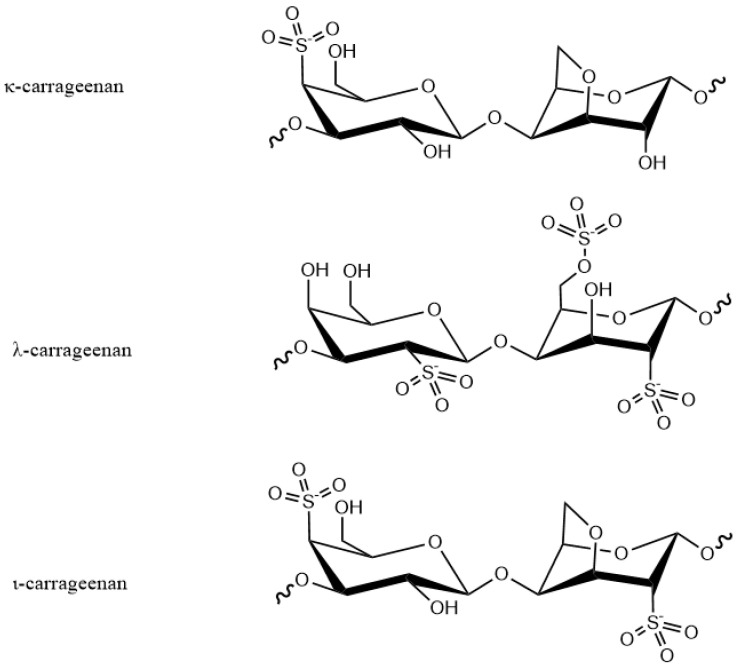
Chemical structures of different types of carrageenans.

**Figure 6 marinedrugs-18-00658-f006:**
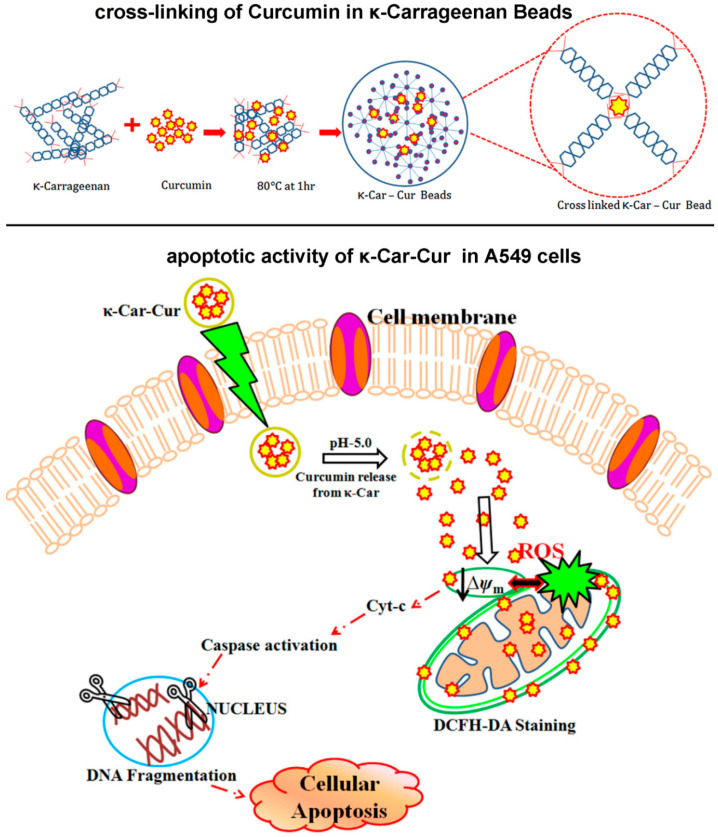
Schematic representation of cross-linking phenomenon of developed κ-Car-Cur materials and its mechanism of action. Reproduced with permission from [45]. Carbohydr. Polym, 2017.

**Figure 7 marinedrugs-18-00658-f007:**
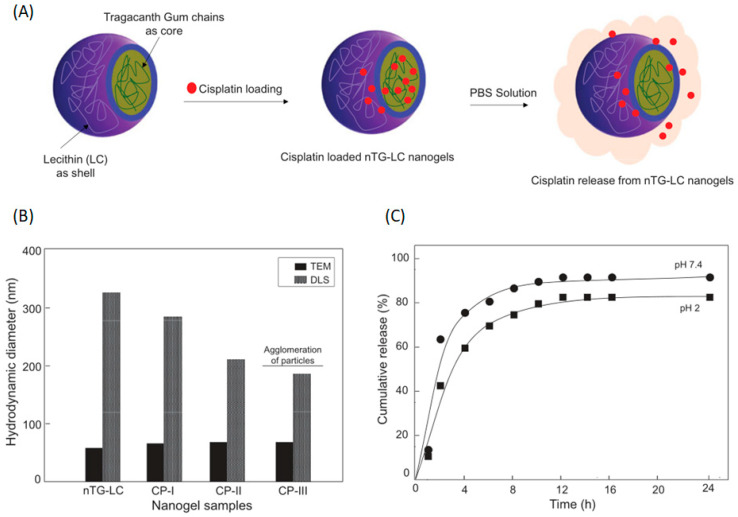
(**A**) Schematic representation for the designing and development of cisplatin (CP) loaded nTG-LC nanogel and their properties. (**B**) Hydrodynamic diameter of nTG-LC and varying concentration of CP loaded nTG-LC nanogel. CP-I, 1.2%, CP-II, 5.2% and CP-III, 8.5%. (**C**) CP release from nTG-LC-CP (5.2% CP) in PBS at different pH 2 and 7.4. Reproduced with permission from [91]. Int. J. Polym. Mater, 2020.

**Table 1 marinedrugs-18-00658-t001:** Biological activities of several typical marine polysaccharides [66,67,68,69,70,71,72,73,74,75,76,77,78,79,80,81,82,83,84,85,86].

Polysaccharides	Biological Functions	Ref
β-glucan	Antioxidant	[66]
	Improves the intestinal microbial system	[67]
	Reduces the risk of cardiovascular disease	[68]
	Reduces blood cholesterol levels	[69]
	Promotes central nervous system axonal regeneration	[70,71]
	Enhances immunity	[72,73,74,75,76,77]
Fucoidan	Antioxidant	[78]
	Anticoagulant	[79,80,81,82,83]
	Anti-inflammatory	[50,84]
	Anticancer	[59,84,85]
	Enhances immunity	[86]
Vulcanized polysaccharides	Enhances immunity	[87]
	Antioxidant	[88]
	Anticancer	[89]

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
