# Peer review of "The Structural Characteristics of Seaweed Polysaccharides and Their Application in Gel Drug Delivery Systems"

_marinedrugs, 2020, doi:10.3390/md18120658_

Round 1

Reviewer 1 Report

The paper by Haowei Zhong et al entitled “The structural Characteristics of Seaweed Polysaccharides and Their Application in Gel Drug Delivery Systems” has been reviewed. The review summarizes the characteristics of alginic acid, carrageenan and other seaweed polysaccharides, and focuses on their application in gel drug delivery systems.

The topic treated by the authors is interesting but, the manuscript has some methodological flaws.

Major suggestions

Please in an appropriate paragraph provide and describe the methodological approach to review the literature cited including inclusion-exclusion criteria, sources etc.

At least the first time in the text and always in the figures, please report algae names according to scientific classification (International Code of Nomenclature for algae, fungi, and plants) avoid the use of the only common name. Make it clear in the text when referring to a species, family, genus etc.

Please review the main text and add where is required the source with the appropriate reference. Some paragraphs lack of adequate bibliographic references (see paragraph 2, 2.1, 2.2)

Figure 2 is too small, bring it vertically and enlarge it

Proof-read the whole text, mistake and misreading are present, below are reported some of them

21-Choose appropriate keywords not present in the title

30-What do you mean with protein amino acids?

47- in vivo and in vitro in italics

48- see the character for preparation

54- is reported: by free radical ROS and ROS. What do you mean?

57- add a reference at the end of the sentence

Table 1 reports species instead of polysaccharides

Etc.

Author Response

Reviewer #1

The paper by Haowei Zhong et al entitled “The structural Characteristics of Seaweed Polysaccharides and Their Application in Gel Drug Delivery Systems” has been reviewed. The review summarizes the characteristics of alginic acid, carrageenan and other seaweed polysaccharides, and focuses on their application in gel drug delivery systems.

Reply: Thanks a lot for the reviewer’s comments.

The topic treated by the authors is interesting but, the manuscript has some methodological flaws.

  1. Please in an appropriate paragraph provide and describe the methodological approach to review the literature cited including inclusion-exclusion criteria, sources etc.

Reply: Thanks for the reviewer’s suggestion. The methodological approach to review the literature cited is added in the revised manuscript. The content is as follows:

“In order to discuss the application of seaweed polysaccharides in gel drug delivery in recent years, we searched the relevant literatures of seaweed polysaccharides based gel drug delivery system in the past three years by website of pubmed, Google academic, and school library electronic journal platform, etc. They were divided into two categories: reviews and articles. From the reviews, we summarized the properties and characteristics of polysaccharides, and traced back the valuable citations. From the articles, the studies focusing on gel drug delivery system are extracted and summarized. Through sorting out the literature information, three types of seaweed polysaccharides: alginic acid and its derivatives, carrageenan and its derivatives, and other seaweed polysaccharides were selected according to the sources of polysaccharides.”

  1. At least the first time in the text and always in the figures, please report algae names according to scientific classification (International Code of Nomenclature for algae, fungi, and plants) avoid the use of the only common name. Make it clear in the text when referring to a species, family, genus etc.

Reply: Thanks to the reviewer for the suggestion, we have revised the algae names according to International Code of Nomenclature for algae, fungi, and plants. "Cyanophyta, Chlorophyta, Xanthophyta, Euglenophyta, Pyrrophyta, Chrysophyta, Charophyta, Rhodophyta, Bacillariophyta, and Phaeophyta" belongs to Latin and meets the requirements of the international nomenclature. However, due to our limited understanding of biological classification, we are unable to make a further detailed description of some examples in this paper. In line with the scientific and rigorous attitude, some examples have been deleted. The revised part is as follows:

(1) “we generally divide marine algae plants into: Cyanophyta, Chlorophyta, Xanthophyta, Euglenophyta, Pyrrophyta, Chrysophyta, Charophyta, Rhodophyta, Phaeophyta. Among these, Rhodophyta, Phaeophyta and Chlorophyta are the most widely used in biomaterials.”

(2) “Alginic acid, also known as acidum alginic, is essentially a straight chain block glucuronic acid, which is widely found in hundreds of Brown algae, Phaeophyta.”

(3) “Carrageenan is a high molecular weight hydrophilic polysaccharide extracted from red algae, Rhodophyta.”

(4) “Fucophyta” have been changed to “Phaeophyta” in Figure 1.

  1. Please review the main text and add where is required the source with the appropriate reference. Some paragraphs lack of adequate bibliographic references (see paragraph 2, 2.1, 2.2)

Reply: Thanks a lot for the reviewer’s comments. We added some references in the revised manuscript. The added references are as follows:

  1. Guo, X.; Wang, Y.; Qin, Y.; Shen, P.; Peng, Q., Structures, properties and application of alginic acid: A review. Int J Biol Macromol 2020, 162, 618-628.
  2. Sun, Y.; Jing, X.; Ma, X.; Feng, Y.; Hu, H., Versatile Types of Polysaccharide-Based Drug Delivery Systems: From Strategic Design to Cancer Therapy. Int J Mol Sci 2020, 21, (23).
  3. S, M.; M, V. K.; Tripathi, A. D.; Ts, R. L., Optimization and characterization of Alginic acid synthesized from a novel strain of Pseudomonas stutzeri. Biotechnol Rep (Amst) 2020, 27, e00517.
  4. Barbosa, A. I.; Coutinho, A. J.; Costa Lima, S. A.; Reis, S., Marine Polysaccharides in Pharmaceutical Applications: Fucoidan and Chitosan as Key Players in the Drug Delivery Match Field. Mar Drugs 2019, 17, (12).
  5. Joshi, S.; Mahadevan, G.; Verma, S.; Valiyaveettil, S., Bioinspired adenine-dopamine immobilized polymer hydrogel adhesives for tissue engineering. Chem Commun (Camb) 2020, 56, (76), 11303-11306.
  6. Vernero, M.; Boano, V.; Ribaldone, D. G.; Pellicano, R.; Astegiano, M., Oral iron supplementation with Feralgine(R) in inflammatory bowel disease: a retrospective observational study. Minerva Gastroenterol Dietol 2019, 65, (3), 200-203.
  7. Quadrado, R. F. N.; Fajardo, A. R., Fast decolorization of azo methyl orange via heterogeneous Fenton and Fenton-like reactions using alginate-Fe(2+)/Fe(3+) films as catalysts. Carbohydr Polym 2017, 177, 443-450.
  8. Zhang, R.; Zhang, X.; Tang, Y.; Mao, J., Composition, isolation, purification and biological activities of Sargassum fusiforme polysaccharides: A review. Carbohydr Polym 2020, 228, 115381.
  9. Sun, H.; Choi, D.; Heo, J.; Jung, S. Y.; Hong, J., Studies on the Drug Loading and Release Profiles of Degradable Chitosan-Based Multilayer Films for Anticancer Treatment. Cancers (Basel) 2020, 12, (3).
  10. Zvukova, N. D.; Klimova, T. P.; Ivanov, R. V.; Ryabev, A. N.; Tsiskarashvili, A. V.; Lozinsky, V. I., Cryostructuring of Polymeric Systems. 52. Properties, Microstructure and an Example of a Potential Biomedical Use of the Wide-Pore Alginate Cryostructurates. Gels 2019, 5, (2).
  11. Porter, G. C.; Schwass, D. R.; Tompkins, G. R.; Bobbala, S. K. R.; Medlicott, N. J.; Meledandri, C. J., AgNP/Alginate Nanocomposite hydrogel for antimicrobial and antibiofilm applications. Carbohydr Polym 2021, 251, 117017.
  12. Wang, Q.; Zhang, L.; Liu, Y.; Zhang, G.; Zhu, P., Characterization and functional assessment of alginate fibers prepared by metal-calcium ion complex coagulation bath. Carbohydr Polym 2020, 232, 115693.
  13. Soulairol, I.; Sanchez-Ballester, N. M.; Aubert, A.; Tarlier, N.; Bataille, B.; Quignard, F.; Sharkawi, T., Evaluation of the super disintegrant functionnalities of alginic acid and calcium alginate for the design of orodispersible mini tablets. Carbohydr Polym 2018, 197, 576-585.
  14. Niu, Y.; Xia, Q.; Li, N.; Wang, Z.; Lucy Yu, L., Gelling and bile acid binding properties of gelatin-alginate gels with interpenetrating polymer networks by double cross-linking. Food Chem 2019, 270, 223-228.
  15. Barbosa, A. I.; Coutinho, A. J.; Lima, S. A. C.; Reis, S., Marine Polysaccharides in Pharmaceutical Applications: Fucoidan and Chitosan as Key Players in the Drug Delivery Match Field. Marine Drugs 2019, 17, (12).
  16. Figure 2 is too small, bring it vertically and enlarge it

Reply: Thanks a lot for the reviewer’s suggestion. Figure 2 has been revised as required.

  1. Proof-read the whole text, mistake and misreading are present, below are reported some of them.

21-Choose appropriate keywords not present in the title

Reply: Thanks a lot for the reviewer’s suggestion. We have changed the key word “seaweed polysaccharide; structural characteristics; gel; drug delivery system” to "alginic acid; carrageenan; seaweed polysaccharides; gels; drug delivery "

30-What do you mean with protein amino acids?

Reply: Thanks a lot for the reviewer’s suggestion. We have changed the “protein amino acids” to “amino acids”.

47- in vivo and in vitro in italics

Reply: Thanks a lot for the reviewer’s suggestion. We have changed the “in vivo and in vitro” to “in vivo and in vitro”

48- see the character for preparation

Reply: Thanks a lot for the reviewer’s suggestion. We have changed the font to be consistent.

54- is reported: by free radical ROS and ROS. What do you mean?

Reply: We have changed the “by free radical ROS and ROS” to “by free radical and ROS”

57- add a reference at the end of the sentence

Reply: Thanks a lot for the reviewer’s suggestion. We have added some references as follows:

  1. Kumar, A.; Buia, M. C.; Palumbo, A.; Mohany, M.; Wadaan, M. A. M.; Hozzein, W. N.; Beemster, G. T. S.; AbdElgawad, H., Ocean acidification affects biological activities of seaweeds: A case study of Sargassum vulgare from Ischia volcanic CO2 vents. Environ Pollut 2020, 259, 113765.
  2. Abdala Diaz, R. T.; Casas Arrojo, V.; Arrojo Agudo, M. A.; Cardenas, C.; Dobretsov, S.; Figueroa, F. L., Immunomodulatory and Antioxidant Activities of Sulfated Polysaccharides from Laminaria ochroleuca, Porphyra umbilicalis, and Gelidium corneum. Mar Biotechnol (NY) 2019, 21, (4), 577-587.
  3. Sajadimajd, S.; Momtaz, S.; Haratipour, P.; El-Senduny, F. F.; Panah, A. I.; Navabi, J.; Soheilikhah, Z.; Farzaei, M. H.; Rahimi, R., Molecular Mechanisms Underlying Cancer Preventive and Therapeutic Potential of Algal Polysaccharides. Curr Pharm Des 2019, 25, (11), 1210-1235.

Table 1 reports species instead of polysaccharide

Reply: Thanks a lot for the reviewer’s suggestion. In fact, “Biological activities of several typical marine polysaccharides” is described in Table 1, so we have changed the “species” to “polysaccharides” in table 1.

Figure 2 should be mentioned in the text

Reply: Thanks a lot for the reviewer’s comments. Figure 2 has been mentioned in the reviesed manuscript. That is:

“The summary of this review is shown in figure 2, and the details will be expanded later.”

Reviewer 2 Report

This review describes interesting topics, and the discussions of this study may well be useful to the scientists in the same field. I believe that the review is well written to publish in Marine drugs.

Author Response

Reply: Thanks a lot for the reviewer’s comments.

Reviewer 3 Report

This article of  Haowei Zhong, Xiaoran Gao, Cui Cheng, Chun Liu, Qiaowen Wang and Xiao Han, entitled “The structural Characteristics of Seaweed Polysaccharides and Their Application in Gel Drug Delivery Systems”, is an interesting review that summarizes the development and  application of seaweed polysaccharides in recent years. This work presents the advantages of alginic acid, carrageenan and other seaweed polysaccharides like fucoidan, β-glucan, Ulvan, GA3P and provides information on their application in gel drug delivery systems.

Τhe manuscript overall is in enough detail written and the English language is correct. The authors have chosen appropriate and up to date references. I recommend the publication of this article after some minor revisions. For example:

Page 2, Line 49: replace “polysaccharides have a good inhibitory effects” with “polysaccharides have good inhibitory effects”

Page 2, Line 54: check this text “reducing the damage induced by free radicals, ROS and ROS”

Page 4, Line 103: replace “meaning the aqueous solution has a certain adhesion properties” with “meaning the aqueous solution has certain adhesion properties”

Page 4, Line 118: replace “which effectively avoids the inactivation of” with “which effectively avoid the inactivation of”

Page 5, Line 150: put a parenthesis after BDC in “and didemethoxycurcumin (BDC”

Page 8, Line 230: replace “is also affected by th presence of cations” with “is also affected by the presence of cations”

Page 8, Line 232: replace “κ- carrageenan exhibit” with “κ- carrageenan exhibits”

Page 9, Line 290: replace “β-glucan is a polysaccharides” with “β-glucan is a polysaccharide”

Page 10, Line 301: replace “has strong oxygen free radical” with “have strong oxygen free radical”

Figure 2 should be mentioned in the text

Author Response

Reply: Thanks a lot for the reviewer’s suggestion. The writing errors have been corrected as required, we have replaced “polysaccharides have a good inhibitory effects” with “polysaccharides have good inhibitory effects”; Replaced “by free radical ROS and ROS” with “by free radical and ROS”; replaced “meaning the aqueous solution has a certain adhesion properties” with “meaning the aqueous solution has certain adhesion properties”; replaced “which effectively avoids the inactivation of” with “which effectively avoid the inactivation of”; replaced “didemethoxycurcumin (BDC” with “didemethoxycurcumin (BDC)”; replaced “is also affected by th presence of cations” with “is also affected by the presence of cations”; replaced “κ- carrageenan exhibit” with “κ- carrageenan exhibits”; replaced “β-glucan is a polysaccharides” with “β-glucan is a polysaccharide”; replaced “has strong oxygen free radical” with “have strong oxygen free radical”.

The modified details are marked in blue in the revise manuscript. We also revise some the English form and correct typos in other places. They are marked with blue color in the revised manuscript.

Round 2

Reviewer 1 Report

The authors made the recommended changes, so overall, I'm positive on this study, even if some names reported in Figure 1 such as Cauliflower, Periwinkle and Marsh Mori still remain ambiguous and do not respect any scientific rigour. I leave the decision on this point to the Editor.

Author Response

Reply: Thanks a lot for the reviewer’s comments. We are very sorry for not checking the manuscript carefully. The names such as Cauliflower, Periwinkle and Marsh Mori, etc. are indeed wrong due to our mistake. We have checked the whole manuscript to avoid these errors. And Figure 1 has been modified in the revised manuscript. Thanks again to the reviewers for such important suggestions. Science is rigorous, we must respect it.
